# *Coccomyxa subellipsoidea* KJ Components Enhance the Expression of Metallothioneins and Th17 Cytokines during Human T Cell Activation

**DOI:** 10.3390/microorganisms12040741

**Published:** 2024-04-05

**Authors:** Toshiro Seki, Shino Ohshima, Satoko Komatsu, Soga Yamada, Hirofumi Kashiwagi, Yumiko Goto, Banri Tsuda, Akiko Kanno, Atsushi Yasuda, Hitoshi Kuno, Noriko M Tsuji, Takashi Shiina, Yoshie Kametani

**Affiliations:** 1Department of Internal Medicine, Division of Nephrology, Endocrinology and Metabolism, Tokai University School of Medicine, Isehara 259-1193, Japan; tsekimdpdd@tokai.ac.jp (T.S.); yasuda1633@yahoo.co.jp (A.Y.); 2Department of Molecular Life Science, Division of Basic Medical Science, Tokai University School of Medicine, Isehara 259-1193, Japan; shino-w@tokai.ac.jp (S.O.); tshiina@tokai.ac.jp (T.S.); 3DENSO CORPORATION, Kariya 448-0029, Japan; satoko.komatsu.j8b@jp.denso.com (S.K.); h.kuno@kj-bio.jp (H.K.); 4Department of Obstetrics and Gynecology, Tokai University School of Medicine, Isehara 259-1193, Japan; 5Department of Palliative Medicine, Tokai University School of Medicine, Isehara 259-1193, Japan; 6Division of Immune Homeostasis, Department of Pathology and Microbiology, Nihon University School of Medicine, Tokyo 113-8602, Japan; 7Division of Microbiology, Department of Pathology and Microbiology, Nihon University School of Medicine, Tokyo 113-8602, Japan; 8Department of Food Science, Jumonji University, Niiza 352-8510, Japan; 9Institute of Advanced Biosciences, Tokai University, Hiratsuka 259-1207, Japan

**Keywords:** *Coccomyxa subellipsoidea* KJ, metallothioneins, Th17 cytokines, T cell activation, immunoregulation, transcriptome analysis, peripheral blood mononuclear cells, inflammatory factors, toxic shock syndrome toxin-1, immune regulatory drugs

## Abstract

*Coccomyxa subellipsoidea* KJ (C-KJ) is a green alga with unique immunoregulatory characteristics. Here, we investigated the mechanism underlying the modification of T cell function by C-KJ components. The water-soluble extract of C-KJ was fractionated into protein (P) and sugar (S) fractions acidic (AS), basic (BS), and neutral (NS). These fractions were used for the treatment of peripheral blood mononuclear cells stimulated with toxic shock syndrome toxin-1. Transcriptome analysis revealed that both P and AS enhanced the expression of the genes encoding metallothionein (MT) family proteins, inflammatory factors, and T helper (Th) 17 cytokine and suppressed that of those encoding Th2 cytokines in stimulated T cells. The kinetics of *MT1* and *MT2A* gene expression showed a transient increase in MT1 and maintenance of MT2A mRNA after T cell stimulation in the presence of AS. The kinetics of Th17-related cytokine secretion in the early period were comparable to those of MT2A mRNA. Furthermore, our findings revealed that static, a STAT-3 inhibitor, significantly suppressed *MT2A* gene expression. These findings suggest that the expression of MTs is involved in the immune regulatory function of C-KJ components, which is partially regulated by Th17 responses, and may help develop innovative immunoregulatory drugs or functional foods.

## 1. Introduction

Microalgae comprising several bioactive compounds, such as polyunsaturated fatty acids, polysaccharides, astaxanthin, and beta-glucan, that exhibit therapeutic potency have garnered increasing attention for their application in various diseases, especially those associated with immune regulation [1,2]. *Coccomyxa* (W. Schmidle, 1901), a green alga that accumulates lipids and metals in the cytoplasm, is a significant contributor to these bioactive compounds and has been implicated in several diseases [3,4]. For instance, monogalactosyl diacylglyceride isolated from *Coccomyxa subellipsoidea* (*Coccomyx* sp.; M. Zuykov 2014) suppresses viral replication in the genital cavity of herpes simplex virus type-2-infected mice [5]. Minerals and metals, including zinc and copper, highly accumulate in the *Coccomyxa* cell bodies and play crucial roles in transcription factors or signaling molecules that are necessary for the immune response [6]. Additionally, the crude polysaccharide AEX isolated from *Coccomyxa gloeobotrydiformis* (H. Reisigl, H 1969) modulated immune responses in chickens [7] and also suppressed lipopolysaccharide (LPS)-induced inflammatory responses in a macrophage cell line (RAW 246.7) [8]. However, despite these reports on the effects of *Coccomyxa* extracts on immune modulation that have been detected mainly using cytokine profiles, their detailed immune cell profiling remains elusive.

Helper T cell subsets, characterized by specific cytokine profiles, play an important role in shifts in immune conditions [9]. Each subset is specifically induced by cytokines released by pathogen-induced immune reactions. Toll-like receptors (TLR) are well-defined pattern recognition receptors involved in the induction of innate and acquired immunity [10,11]. They signal through specific ligands in different microbes and recognize pathogen-associated molecular patterns, such as LPS and CpGs, to induce innate immune responses. *C. gloeobotrydiformis* contains AEX, lipids, and unidentified unique molecules; however, lipid-based molecules such as LPS are prone to abortion during the purification process of water-soluble materials. Moreover, the polysaccharides derived from *C. gloeobotrydiformis* suppress LPS-induced inflammation by modulating the regulation of various signaling pathways [8]. In our previous study, we showed that C-KJ extracts induce the differentiation of highly competent stem cell-like memory T cells (T_scm_) [12] and suppress superantigen-induced immune responses in vitro [13]. Additionally, in a clinical study, C-KJ supplementation suppressed the deterioration of physical conditions related to immune and neuronal functions in healthy adults [14]. These studies indicate that C-KJ components do not function as ligands of TLR, suggesting the existence of a new molecular mechanism associated with the modulation of the immune system for T cell regulation by *Coccomyxa* components. 

Furthermore, sterols also regulate innate and acquired immunity [15,16]. The sterols contained in *Coccomyxa* are similar to those in evolutionarily higher plants, and no specific sterols or steroids have been identified in this genus [17]. Therefore, sterols may not be major players in imparting the unique functions of C-KJ extracts. In contrast, *Coccomyxa* as represented by *Coccomyxa actinabiotis* sp. nov (C. Rivassesu 2016) contains high amounts of minerals such as zinc and copper [4]. Among these low molecular weight molecules, zinc plays a crucial role in maintaining homeostasis of the immune system [18,19]. Many transcription factors are zinc-finger proteins, and defects in these proteins can lead to serious immune dysfunction [20]. Therefore, enhancing the expression of genes involved in mineral regulation may contribute to addressing these issues. Because transcriptome analysis has been demonstrated as a powerful tool to unravel the intricate molecular interactions involved in immune regulation, it might provide insights into the molecules that function as key factors in the regulation of the immune system, such as minerals like zinc that are often linked to larger molecules like glucosides or protein complexes. 

Based on these previous findings, we hypothesize a novel molecular mechanism associated with the modulation of the immune system by *Coccomyxa* components, particularly in T cell regulation. To test this hypothesis, in this study, we aimed to investigate the gene expression profile of T cells induced by water-soluble *Coccomyxa* extract components, using microarray and real-time PCR techniques together with detailed flow cytometry (FCM) analysis. The findings of this study unveil the potential dual regulation by meal-, mineral-, and interleukin (IL)-17-related pathways, shedding light on unexplored aspects of immune modulation by C-KJ components.

## 2. Materials and Methods

### 2.1. Ethical Approval 

Human peripheral blood mononuclear cells (PBMCs) were derived from healthy donors (HDs) after receiving written informed consent from the participants. The study was approved by the Institutional Review Board of the Tokai University Human Research Committee (approval no. 20R051, 21R059) and conducted in accordance with the guidelines of the Declaration of Helsinki and the Japanese federal regulations outlined for the protection of human participants. Healthy donors without a history of malignant diseases were selected to obtain blood samples for this study.

### 2.2. Preparation of Human PBMCs

RPMI 1640 medium and supplements were purchased from Nissui, Co., Ltd., (Tokyo, Japan) and peripheral blood (PB; 50 mL) was collected from each HD in the morning using Vacutainer ACD tubes (NIPRO Corporation, Osaka, Japan) containing heparin. The collected PB was immediately transferred to a 10-mL Ficoll–Hypaque density gradient medium (Sigma-Aldrich, London, UK) and centrifuged (500× *g*, 30 min, 20 °C) to isolate mononuclear cells. The remaining erythrocytes were removed by osmotic lysis. The cells were washed with phosphate-buffered saline (PBS) for 5 min at 300× *g*, 4 °C, and the cell number was counted.

### 2.3. Preparation of C-KJ Fractions

The strain *Coccomyxa* sp. KJ (C-KJ; IPOD FERM BP-22254) was provided by Denso CORPORATION, Kariya, Japan). 

As shown in Appendix A, the water-soluble extract of C-KJ was divided into two parts. One part was subjected to ultra-filtration to obtain the low molecular weight (MW; ≤3 kDa) compound (LWCO), and the other part was used for isolation of the protein (P) and sugar (S) fractions. The protocol for isolation, purification, and quantification of P fractions in C-KJ extracts is outlined in Appendix A. 

Lyophilized C-KJ sample (2.5 g) was added to 25 mL of distilled water and incubated with shaking at 37 °C and 100 rev/min for 6 h followed by precipitation with 5 mg/mL polyethyleneimine (final concentration; 0.05%). The suspension was centrifuged at 3600× *g* for 10 min, and the supernatant was salt-precipitated until 80% saturation using ammonium sulfate. The precipitates were desalinated and applied to a Hitrap DEAE FF column equilibrated with 10 mM phosphate buffer (pH 7.0); the proteins were eluted using a 10–1000-mM NaCl gradient followed by sodium dodecyl-polyacrylamide gel electrophoresis profiling of the peaks. The eluents amplifying the protein bands were pooled and used as the P fraction. The supernatants of ammonium sulfate were desalted, concentrated, and precipitated using 80% (*v*/*v*) ethanol. The precipitate was dissolved in water to obtain the water-soluble S fraction. The S fraction was further fractionated using anion exchange chromatography and divided into acidic (AS), neutral (NS), and basic sugar (BS) fractions. Briefly, the S fraction was subjected to a DEAE-650M column (TOYOPEARL, 40 mL column volume, Tosoh, Tokyo, Japan), washed with distilled water, and eluted with two column volumes of 100-, 200-, 300-, 400-, 500-, and 1000-mM NaCl solutions. The presence of sugar was detected by the phenol sulfuric acid method, and the fractions containing sugar were collected and dialyzed in distilled water with a 3.5-kDa membrane. The dialyzed product was concentrated using an evaporator to obtain BS. The column chromatography of the flow-through fraction using a CM-650 column (TOYOPEARL, 40 mL column volume) followed by washing with distilled water and elution with the above-mentioned concentrations of NaCl solutions yielded AS. Subsequently, the flow-through of the CM columns was dialyzed and concentrated, as described above, to obtain NS.

### 2.4. Culture of Human PBMCs

The cells were seeded in 6-well plates and cultured at a density of 1 × 10^6^ cells/mL in RPMI 1640 medium (Nissui, Co., Ltd.) containing 10% fetal calf serum (Sigma Aldrich, Bloomberg, MO, USA) and antibiotics (streptomycin, 0.1 mg/mL; penicillin 100 unit/mL; Meiji Seika, Tokyo, Japan) in the presence of 1 µg/mL toxic shock syndrome toxin-1 (TSST-1; Toxin Tec. Sarasota, FL, USA) at 37 °C and 5% CO_2_. The cells were incubated with varying concentrations of C-KJ fractions in culture medium, collected at 72 h, washed with PBS, and stained with fluorochrome-labeled monoclonal antibodies (mAbs) for FCM analysis. In the presence of cortisol, the cells were cultured for 72 and 216 h, and the hydrocortisone concentration was 1 μM. Signal transducer and activator of transcription 3 (STAT3) inhibition assay was performed using 1 μM Sttatic (Axon Medchem, Reston, VA, USA). After 72 h, the supernatant and cells were collected, and RNA was extracted from the cells. 

### 2.5. Analysis of Immune Cell Composition Using FCM

Mononuclear cells were collected from each well, quantified, and stained with appropriate dilutions of fluorochrome-labeled mAbs for 15 min at 4 °C, followed by washing with 1% (*w*/*v*) bovine serum albumin (Sigma Aldrich) in PBS. Cells were analyzed for the surface expression of differentiation antigens using a BD LSRFortessa^TM^ flow cytometer (BD Biosciences, Franklin Lakes, NJ, USA). For each analysis, living white blood cells or lymphocytes were gated for propidium iodide and analyzed using the FlowJo software v10.3 (BD Biosciences, San Jose, CA, USA). The mAbs used for staining are listed in Appendix A. 

### 2.6. Quantification of Cytokines Secreted by Cultured PBMCs

Supernatants of the cultured cells were collected for cytokine quantitation using bead-based multiplex LEGENDplex (BioLegend, San Diego, CA, USA) according to the manufacturer’s instructions. Briefly, 25 µL of supernatant was mixed with 25 µL of capture beads and incubated for 2 h at 25 °C. The beads were then washed, mixed with detection antibodies, and incubated for 1 h at room temperature. Subsequently, streptavidin–phycoerythrin was added, and the mixture was incubated for 30 min at room temperature. Finally, the beads were washed and analyzed by FCM. The cytokines IL-1β, interferon (IFN)-α, IFN-γ, tumor necrosis factor-α (TNF-α), monocyte chemotactic protein (MCP)-1, IL-2, IL-4, IL-5, IL-6, IL-8, IL-9, IL-10, IL-12p70, IL-13, IL-17A, IL-17F, IL-18, IL-22, IL-23, and IL-33 were quantified. Analysis was performed using the BD FACSVerse^TM^ Flow Cytometer (BD Biosciences). The data were analyzed in picogram per milliliter using LEGENDPlex™ V8.0 (BioLegend). IFN-γ was quantified using ELISA with serially diluted samples using OptEIA^TM^ Human IFN-γ ELISA Set (BD Biosciences) according to the manufacturer’s instructions.

### 2.7. Purification of T Cells

A Pan T Cell Isolation Kit (Miltenyi Biotec, Bergisch Gladbach, Germany) was used for T cell sorting. PBMCs were cultured as described previously for 72 h. Briefly, the cells were collected, washed, and incubated with the Pan T-cell biotin–antibody cocktail at 4 °C for 5 min. Afterward, 40 µL of wash buffer followed by 20 µL of ion Pan T cell microbead cocktail were added and incubated at 4 °C for 10 min. The T cells were then sorted using the Automacs system (program: depletion; Miltenyi Biotec) and labeled using CellTrace™ Cell proliferation kits (Thermo Fisher Scientific, Waltham, MA, USA), following the manufacturer’s instructions. 

### 2.8. Microarray and Quantitative Real-Time PCR (qRT-PCR) Analysis

Cells were stored in TRIzol (Invitrogen, Carlsbad, CA, USA) at −80 °C and extracted according to the manufacturer’s instructions. RNA concentration was determined by measuring the absorbance at 260 nm using a NanoDrop1000 (Thermo Fisher Scientific, Rockford, IL, USA). Purity was estimated using a relative ratio of 260/280 nm, and integrity was checked by agarose gel electrophoresis. The samples with >95% purity and integrity were used for further analysis. The cDNAs were synthesized from total RNA (2 μg) using a High-Capacity cDNA Reverse Transcription Kit (Life Technologies, Burlington, CA, USA). Microarray analysis on a 3D gene-DNA chip was outsourced to Toray Industries, Inc. (Tokyo, Japan). 3D gene measurements were performed in triplicate (n = 3). 

TaqMan probes for human metallothionein 1 and 2 (ID Hs01591333_g1, Hs00831826_s1, Thermo Fisher Scientific, Co., Ltd., Waltham, MA, USA) were used, and qRT-PCR was conducted using Applied Biosystems StepOnePlus Real-Time PCR Systems. Commercially available TaqMan Fast Universal PCR Master Mix (Thermo Fisher Scientific, Rockford, IL, USA) was used for PCR amplification and detection. All samples were analyzed in triplicate (n = 3). 

### 2.9. Zinc Quantification

Lyophilized samples were subjected to thermal decomposition and concentration in the presence of sulfuric acid and nitric acid and dissolved in nitric acid. Zinc and copper were quantified using Inductively Coupled Plasma Mass Spectrometry (Torey Research Center, Tokyo, Japan).

### 2.10. Data Analysis

The data were imported into the software program GeneSpring GX14.9.1 (Agilent Technologies, Santa Clara, CA, USA) for filtering and basic statistical analysis. Differentially expressed genes (DEGs) in TSST-1, TSST-1 (P), and TSST-1 (AS) were identified with a cut-off of the Benjamini–Hochberg adjusted *p*-value below 0.05, and those with a fold change of at least 2.0 according to Welch’s *t*-test were considered significantly differentially expressed. Unsupervised hierarchical clustering and principal component analyses were performed to visualize the overall expression characteristics of all samples used in this study. Biological function and pathway analyses were performed using the online Database for Annotation, Visualization, and Integrated Discovery (DAVID) toolkit 6.8, which is an ontology-based web tool (https://david.ncifcrf.gov/home.jsp accessed on 1 June 2023). Gene lists defined as DEG for each group were uploaded using official gene symbols to identify enriched gene ontologies for gene expression and functional pathway analyses. The biological functions of the selected genes were analyzed using the Gene Ontology (GO) database [21] and the Kyoto Encyclopedia of Genes and Genomics (KEGG) [22]. Protein–protein interactions and gene function prediction analyses were performed using GeneMANIA (http://genemania.org/ accessed on 7 November 2023).

### 2.11. Statistical Analyses

For real-time PCR, LEGENDplex and volcano plots, statistical analyses were performed using one-way repeated ANOVA and paired Student’s *t*-test (Microsoft Excel ver.16.83) (Microsoft, Redmond, WA, USA). Data are presented as means ± standard deviation.

## 3. Results

### 3.1. C-KJ Fractionation

In the process of C-KJ fractionation, the P fraction was obtained by ammonium sulfate precipitation and desalination. Hitrap DEAE FF column chromatography patterns of the extract revealed three peaks, among which the second peak was the major one (Appendix A). Sodium dodecyl-sulfate polyacrylamide gel electrophoresis of the extracts detected proteins with molecular weights ranging from 52–72 kDa only for the eluents containing the second peak. Among the amplified bands, the one at 26 kDa was the major one (Appendix A). Moreover, all eluents of the second peak analyzed showed similar amplification patterns; therefore, they were pooled and used as the P fraction. The supernatants of ammonium sulfate precipitation were subjected to desalination followed by 80% ethanol precipitation to obtain the water-soluble S fraction (the precipitate). Subsequently, the supernatant obtained in the ethanol precipitation process was dried after removing the solvents to obtain the intermediate-MW compound (Int; Appendix A). The final yields of P, S, and Int compounds were 2, 234, and 780 mg, respectively. The yields of the fractions AS, NS, and BS obtained by fractionation of BS were 91 (*ca.* 50 mL); 490 (*ca.* 50 mL); and 2770 μg/mL (*ca.* 30 mL), respectively. 

### 3.2. Characterization of T Cells Differentiated in the Presence of C-KJ Fraction

The PBMCs were stimulated with TSST-1, an enterotoxin of *Staphylococcus aureus* (Ogston 1884) in the presence of crude extract (CE), P, S, and Int (at an equivalent of 300 μg/mL of CE). We divided these into small lymphocyte gates, indicating non-activated cells, and large lymphocyte gates, indicating activated cells, which are shown in Appendix A. The analysis revealed that most surviving human cells in each fraction after TSST-1 stimulation were T cells, and large lymphocyte-gated cells, which expressed the activation markers CD25 and PD-1, were increased in both CD4^+^ and CD8^+^ T-cells (Appendix A). We compared the activation and differentiation markers of these T cells in detail. TSST-1-stimulated cells in the presence of P [TSST-1 (P)] and S [TSST-1 (S)] increased the ratio of CD45RA^+^CD62L^+^CD95^+^ T_scm_, which was comparable with that of CE, while TSST-1 (Int) did not increase CD45RA^+^CD62L^+^CD95^+^ T_scm_ compared with TSST-1 alone (TSST-1) (Figure 1a and Appendix A). Therefore, we selected P and S as immune-modulatory fractions and performed further analyses. After further fractionation of S, as shown in Appendix A, we compared AS, NS, and BS effects on T cell activation (Appendix A). No obvious difference was observed in the ratio of activated T cells among the sugar fractions, but TSST-1 (AS)-stimulated cells showed an increase in the T_scm_ ratio and the highest CD95 mean fluorescent intensity, while those in the cells treated with NS or BS showed no differences compared to TSST-1 (Figure 1b and Appendix A). Among the other T_scm_ markers, the expression of CD127 was increased in the large lymphocyte-gated T_scm_ fraction (Figure 1c–g). T_scm_ is a T cell subset with high potential to produce competent T cells. Our findings suggest that AS, containing the highest ratio of T_scm_, induces and regulates T cell activation and differentiation; therefore, we selected AS and P for further analyses.

Subsequent comparative analysis of the effects of the P and AS fractions on cytokine production revealed that both treatments significantly decreased the production of TNF-α, IL-2, IL-5, IL-13, and IL-4 but increased that of IL-17A and IFN-γ compared with TSST-1 treatment alone (Figure 2). The levels of IL-10 tended to increase with the treatments, whereas those of the other cytokines were not affected. These results suggest that both P and AS affect T cell differentiation. The AS and P fractions decreased the production of TNF-α and Th2 cytokines but increased that of IFN-γ and Th17A compared to TSST-1 alone. However, no significant differences among the production levels of these cytokines between AS and P were observed.

### 3.3. Changes in Global Gene Expression in the Presence of C-KJ Components

Microarray analyses and comparison of the transcriptome profiles of T cells stimulated with TSST-1 in the presence of C-KJ (P) and (AS) components identified DEGs showing >2-fold change in expression. A total of 119 genes were upregulated and 297 were downregulated in TSST-1 (P) vs. TSST-1, whereas 198 genes were upregulated and 288 were downregulated in TSST-1 (AS) vs. TSST-1. Among these, 51 upregulated and 102 downregulated genes were common (Figure 3a,b). As shown in Figure 3c, hierarchical clustering analyses revealed that the global gene expression patterns of TSST-1 (P) and TSST-1 (AS) were similar to each other but different from that of TSST-1. 

GO analysis of upregulated DEGs revealed enrichment of two categories: one enriched with terms such as cellular responses to metal ions (e.g., copper, cadmium, and zinc ions), while the other harbored terms such as inflammatory and immune responses, involving positive regulation of cytokine production (Figure 4a). For these enriched terms, no significant differences were observed between the TSST-1 (P) and TSST-1 (AS) groups. GO analysis of downregulated DEGs did not identify any enriched term.

KEGG analysis of upregulated DEGs revealed enrichment of the IL-17 signaling pathway, cytokine–cytokine receptor interaction, rheumatoid arthritis, mineral absorption, NF-kappa B signaling pathway, and lipid and atherosclerosis (Figure 4b). Most of these involved immune-related DEGs. TSST-1 (P) enriched immune-related DEGs to a greater extent than TSST-1 (AS). 

Next, we identified DEGs whose expression levels differed among TSST-1, TSST-1 (P), and TSST-1 (AS) treatments using a volcano plot. Both TSST-1 (P) and TSST-1 (AS) significantly increased the expression of *MT* family genes (Appendix A and Appendix A). In contrast, the levels of Th2-related cytokines, IL-5 and CD80, co-receptors of human leukocyte antigens, were decreased. Moreover, DEGs between TSST-1 (P) and TSST-1 (AS) tended to differ; however, no clear categories were observed (Appendix A). These results suggest that both C-KJ extract fractions enhanced the expression of metal ion-related MTs and immune response-related genes, especially IL-17-related genes. The Th2 response tended to decrease with both P and AS treatment. 

Next, we analyzed the relationship between DEGs using functional enrichment analysis of the protein–protein interaction networks using GENEmania. Both TSST-1 (P) and TSST-1 (AS) formed two large clusters (Figure 4c,d)—one cluster contained *MT* genes and the other contained immune-related genes. The *MT* cluster contained *MT1*, *MT2*, *MT3*, and other isotypes; however, molecules related to the uptake of metal ions, named *ZIP* family genes, were not detected. The immune-related cluster contained chemokines and inflammatory cytokines such as *IL-6* and *Th17*. *IL-6* positively correlates with *MT* expression, while IL17 has been reported to be suppressed by MTs [23]. As for *IL-1β*, the antagonist *IL-1RN* also slightly increased. However, a strong relationship between *MT* genes and immune-related genes was not observed in STRING analysis. Chemokines such as *CXCL2*, *CXCl8*, and *CXCL3* were commonly increased, and other chemokines and related genes were strongly correlated with inflammatory cytokines.

Because *MT* expression was increased in the presence of the P or AS fractions, we examined whether these components involve metal ions, such as zinc and copper. Ultra-filtration of the soluble fraction five times and measurement of zinc and copper using an Agilent 8800 yielded 12 mg/mL zinc and 35 mg/mL copper. These results suggest that the metals were not free but bound to proteins or glycosides (Appendix A). Together, these findings indicate that P and AS contain zinc and copper and strongly induce MTs and immune-related genes, especially the Th17 response.

### 3.4. MTs Are Differentially Expressed upon T Cell Stimulation in the Presence of C-KJ Fractions

To confirm the induction of *MTs* and immune-related genes by the P and AS fractions, we further quantified the expression of *MT1* and *MT2* in the presence of TSST-1 (P) and TSST-1 (AS) using qRT-PCR. TSST-1 stimulation significantly induced *MT1A* expression, which was further increased in the presence of P and AS (Figure 5a). Conversely, *MT2A* expression was significantly decreased with TSST-1 stimulation alone but increased by TSST-1 stimulation in the presence of P and AS (Figure 5b); however, the expression of *MT1A* and *MT2A* did not differ significantly between the P and AS treatments. Because the levels of expression of *MT* genes were similar between P and AS, we next examined the kinetics of *MT* expression using AS as the C-KJ component. Additionally, we compared the effect of AS with that of glucocorticoids (cortisol; COR), which suppress T cell activation. *MT1A* expression significantly increased with AS on day 3 and decreased on day 6 to a level similar to that of the control. However, after TSST-1 stimulation, *MT2A* levels decreased on day 3 and then increased on day 6, whereas they increased until day 3 and then remained stable in AS treated cells. The expression of both *MTs* in COR-treated cells tended to be lower than those in the control (*p* > 0.05).

Next, we examined the effects of AS and COR on the kinetics of cytokine production by T cells stimulated with TSST-1. The expression of most cytokines increased on day 3 and decreased on day 6 (Figure 6). In (AS)-treated cells, TNF-α, IL-2, IL-5, and IL-13 levels were decreased, but those of IFN-γ, IL-17, and IL-10 increased. The levels of IL-10 were similar in AS- and COR-treated cells. The expression of IFN-γ, which showed a similar kinetic to the mRNA of *MT1A*, was the highest on day 3. The IL-9 level was enhanced on day 6, whereas the IL-22 level tended to increase. Th17 cytokines (IL-17A, IL-17F, and IL22) were not downregulated on day 6; however, their expression levels were either maintained or increased, and the kinetics were similar to those of *MT2A* expression. 

These results suggest that C-KJ components increased the expression of *MT1A* and *MT2A*. However, the kinetics differed; while the IFN-γ expression kinetics were similar to those of *MT1A*, the IL-17 expression kinetics were similar to those of *MT2A*. The cytokine profile was changed by the C-KJ components, and this was not strictly biased to one subset. However, the increase in IFN-γ and IL-17 suggests that the profile was balanced and adequate to block both bacteria and virus infection.

### 3.5. MT Expression Is Partially Controlled by STAT-3

Zinc enhances immune function; nevertheless, it suppresses Th17 cell function in a dose-dependent manner [23,24]. IL-6 induces IL-17 production, through which *MT* expression is enhanced and negative feedback occurs. However, our results showed an increase in the protein levels of IL-17 and IL-6, and the mRNA levels of *MTs* in the presence of either P or AS. Therefore, we aimed to clarify how IL-17 expression affects MT expression using static, a small molecule inhibitor of STAT-3.

*MT2A* expression was significantly suppressed; no change was observed in *MT1A* expression by static stimulation after TSST-1 stimulation (Figure 7a). The Th17-related cytokine IL-22 was significantly decreased, although IL-17 secretion did not change (Figure 7b). IFN-α also decreased, and IL-1β tended to decrease, while IL-2 significantly increased, suggesting that the cytokine profile changed from an innate inflammation response to an adaptive immune response.

## 4. Discussion

Studies on immune regulation by green algal components, including those from *Coccomyxa*, are limited [2,5,6,7,25,26]. Herein, we undertook a comprehensive investigation into the immune regulatory properties of C-KJ extracts, specifically focusing on the fractionation process, T cell characterization, and changes in gene expression. Our findings reveal a multifaceted effect of C-KJ components on the immune system, with notable outcomes in T cell activation and the expression of key genes. The fractionation of C-KJ extracts allowed us to pinpoint the unique properties of P and AS fractions, shedding light on their distinct roles in modulating the immune response. Both fractions demonstrated similar effects on T cell activation and the expression of surface markers related to T_scm_, underscoring the significance of non-protein factors in immune regulation.

Consistent with our previous study [13], here we showed that both P and AS treatment decreased IL-2 and increased IL-17 expression. Conversely, IL-5 and IL-13 production decreased upon P and AS treatment, which differs from the results of our previous study [13]. This discrepancy between results could be because we used partially purified P and AS in this study, which may have depleted some Th2 cytokine-inducing factors. Another study has shown that the water extract of *Chlorella sorokiniana* enhanced T cell secretion of Th1 cytokines and concomitantly enhanced TNF-α [27], which was decreased by C-KJ extract. Nonetheless, the secretion of inflammatory cytokines, such as IL-6, IL-1β, and TNF-α, has been shown to decrease in RAW 246.7 macrophage cell lines upon treatment with *C. gloeobotrydiformis* extracts [8]. Taken together, these studies suggest the presence of various unique immune modulators in the water-soluble components of green algae. 

As the crude extract of C-KJ was heat-treated, which presumably may have denatured almost all proteins, we speculated that proteins themselves may not bear the functionality of immune system remodeling. Therefore, we focused on investigating the effect of C-JK extracts on T cell activation and differentiation to identify new immune regulatory factors, especially those related to glycosides. 

To this end, we purified T cells and performed microarray analysis, which revealed that AS and P exhibited highly similar gene profiles for immune regulation in the GO, KEGG, and STRING analyses. These findings suggest that the components affecting immune regulation are glycosides or other low molecular weight molecules bound to larger molecules. In the analyses, P- and AS-treated T cells expressed a significantly higher number of genes related to pathways involving immune regulation. P and AS demonstrated a dual role in immune modulation—they decreased Th2 cytokines together with class II MHC [28], suggesting a potential regulatory role of these components in the regulation of allergic inflammation [29]. Moreover, P and AS partially regulated the Th1 response by suppressing IL-2 production. Moreover, the level of IL-9, categorized as Th9, exhibited an upward trend, suggesting a possible enhancement of the Th17 response [30] by secreting IL-21 [31]. Th17 cells, a subset of helper T cells, play a crucial role in inducing neutrophil mobilization and exerting anti-pathogenic effects [32]. Therefore, the Th17 response might contribute partially to the observed effects of water-soluble *Coccomyxa* extract in our clinical study [14]. Moreover, P and AS enhanced the expression of chemokine genes, including *CXCL2* and *CXCL8*. The increased secretion of IFN-γ and IL-10 suggests the induction of a non-biased immune response other than C-KJ component-induced Th2 reduction. These findings collectively underscore the multi-faceted impact of P and AS fractions on immune regulation, implicating their potential significance in therapeutic interventions and immune-related studies.

*Coccomyxa* has been recognized as an alga that accumulates high levels of metals and minerals [4,33]. In this study, we identified a prominent gene category associated with metal and mineral ion-related genes, including zinc and copper, which exhibited an increase in P and AS fractions. Specifically, genes of the MT family that bind to metal and mineral ions, mainly zinc and copper, are highly expressed [34,35,36]. MT1 and MT2 are involved in immune regulation, while MT3 is usually involved in the neural system [37]. Usually, the zinc signal mediated via zinc transporters [38] enhances the expression of MTF-1 transcription factor, leading to the transcription of *MT*. These studies suggest that the abundant mineral and metal content in C-KJ components may induce *MT* expression. However, we did not observe any increase in these transporters, suggesting the existence of a specific pathway induced by C-KJ components that merits further exploration. 

The MT–zinc axis is closely associated with immune regulation in infectious diseases or atopies [29,39,40]. MT1 and MT2 are two major MTs involved in immune function [35]; however, the distinct roles of MT1 and MT2 have not been clarified. T-bet, the master regulator of the Th1 subset, requires zinc as a cofactor, and defect in T-bet suppresses Th1 subset differentiation [41]. MT1 shifts the differentiation of Th cells towards Treg cells [23] and downregulates MHC-II and IL-1β, IL-6, IL-12, and TNF-α on the dendritic cells [28,42]. A previous study has reported that immune system activation possibly involves the induction of MT by TSST-1; however, TSST-1 did not affect IL-1 or TNF-α levels but enhanced that of IL-6 in mouse livers. Moreover, a study showed that IL-6 induction preceded MT mRNA induction in normal-cytokine-producing mice but not in low-cytokine-producing mice [43]. Therefore, that study could not reach a consensus on whether MT induces IL-6 followed by IL-17 stimulation or not. With respect to chemokine gene expression, under MT1/MT2 deficient conditions, the number of circulating lymphocytes decreases, suggesting that MT is a chemoattractant [44]. However, in this study, we did not find any strong correlation between MT and chemokines in the protein–protein interaction network analysis.

Therefore, we further examined *MT1* and *MT2* expression to explore the relationship between the expression of MT and immune-related genes. *MT1* expression was transiently enhanced after TSST-1 stimulation, whereas *MT2* expression was downregulated. AS enhanced *MT1* expression and maintained *MT2A* expression. Moreover, *MT2A* expression was partially affected by STAT-3, which enhanced IL-17 expression, indicating that *MT2A* expression was induced in conjunction with an IL-17-related inflammatory response. However, this finding contradicts those of previous studies demonstrating that IL-17 expression is suppressed by MTs or zinc [28,45]. Moreover, the response is not equivalent to the immune suppression caused by zinc itself, which specifically suppresses Th2 cytokines and increases Th1 cytokines [20]. Similarly, it does not align with the response to COR, as both MT1 and MT2A expression decreased under COR treatment. Moreover, KEGG pathway analysis revealed a more prominent immune-related gene expression in the P-treatment group, suggesting that the P fraction involves more immune modification factors. Because the P and AS fractions were dialyzed and almost all free metal ions were speculated to be depleted, the zinc and copper ions detected in these fractions suggested their existence as glycosides or metal-binding proteins. Together, these findings suggest that the effect of the C-KJ component is likely induced by zinc- or copper-related glycoside or the derivative(s). 

Because of this complexity, the relationship between the enhanced immune response, specifically cytokine production, and MT or zinc remains elusive. However, inflammatory cytokines have been shown to induce upregulation of ZIP-8, a zinc-importer, and the uptake of zinc enhances IL-6 production, suggesting a feedback loop between inflammation and zinc uptake [46]. Because C-KJ involves zinc, the increased zinc uptake could potentially induce or maintain MT expression in T cells, which, in turn, might further enhance IL-17 or IL-22 production through elevated IL-6 levels. However, inflammation induced by C-KJ components may lead to concurrent zinc uptake and MT expression, or the unique cytokine profile associated with inflammation may contribute to enhanced MT expression. A comprehensive analysis is imperative to elucidate the mechanism underlying C-KJ components. 

While our study provides valuable insights, it is crucial to acknowledge its limitations. Firstly, in the analyses of immune reaction, we used TSST-1 to stimulate human T cells. Because TSST-1 is a superantigen which crosslinks multiple TCRβ alleles and major histocompatibility complex antigens [47], the T cell response is not antigen-specific. Therefore, TSST-1 stimulation in this study does not exactly mimic the antigen-specific response of T cells. Moreover, in this study, we used PBMCs for T cell stimulation, which contained not only naïve but also effector and memory T cells, although the proportion was not high. Therefore, in future, we need to purify naïve T cells and stimulate by CD3 and CD28 crosslinking to clarify if the same change of gene expression occurs in the typical antigen-specific response. Secondly, as for the C-KJ extracts, our focus was on the fractionated components of C-KJ, highlighting the need for further study to purify and identify the active principles of these extracts. Additionally, it is essential to conduct further investigations using in-vivo models to validate and extend our findings.

## 5. Conclusions

Our investigation into the immunoregulatory properties of P and AS fractions of water-soluble C-KJ extract unveiled a distinctive effect on T cell activation and immunomodulatory gene expression profiles. The increased expression of MTs, particularly MT1 and MT2, indicated their significant role, possibly influenced by interactions with zinc and copper. The unique features of the P and AS fractions, emphasizing the contribution of non-protein factors, were evident in comparable immune regulation profiles. Our study sheds light on Th17-related inflammatory responses, cytokine production, and T cell differentiation, with complex effects on Th1 and Th2 responses. The intricate interplay between C-KJ components, immune cells, and cytokine networks, coupled with the regulation of MT expression, suggests a potential feedback loop deserving further exploration. Overall, the upregulation of MT- and immune-related genes by C-KJ components presents promising avenues for developing novel immunoregulatory interventions and functional foods.

## Figures and Tables

**Figure 1 microorganisms-12-00741-f001:**
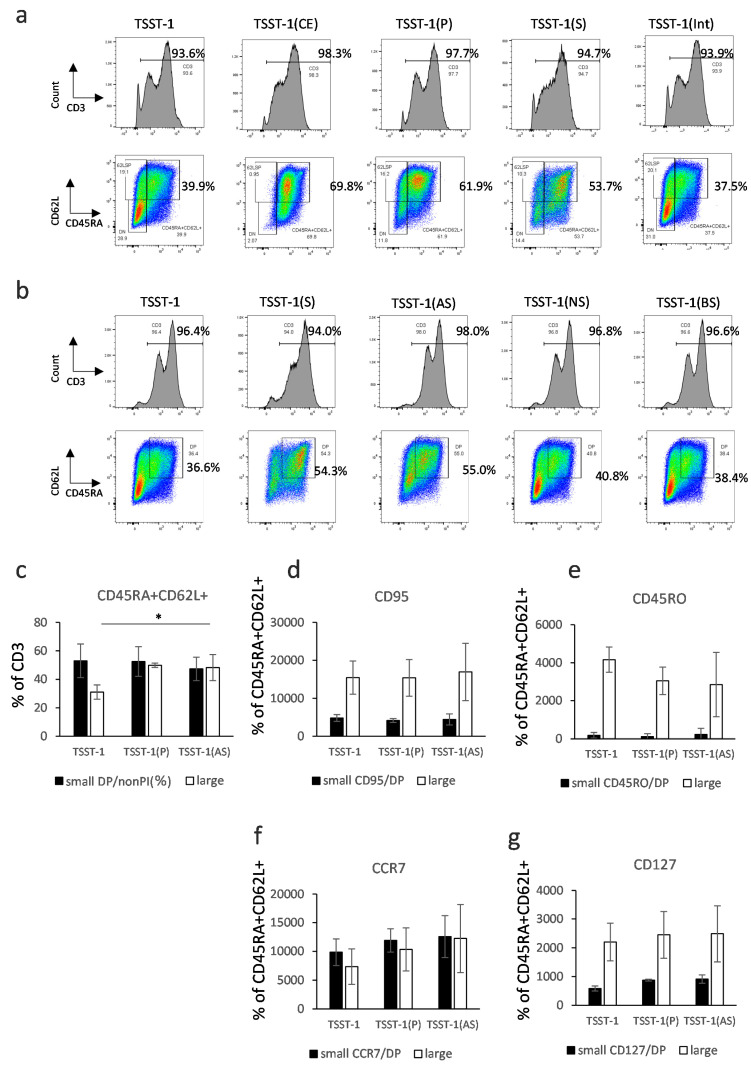
Effect of the protein and sugar sub-fractions of C-KJ on T cell differentiation. (**a**) CD3 expression (histograms) and CD45RA/CD62L expression on CD3^+^ T cells (quadrants). TSST-1, TSST-1 treatment; TSST-1 (CE), TSST-1 + crude extract; TSST-1 (P); TSST-1 + protein fraction; TSST-1 (S), TSST-1 + sugar fraction; TSST-1 (Int), TSST-1 + intermediate molecular fraction. The percentages of CD3^+^ and CD45RA/CD62L + (DP) cells are shown. (**b**) The same expression in the T cells stimulated with partially purified C-KJ components. TSST-1 (AS); TSST-1 + acidic sugar fraction, TSST-1 (NS); TSST-1 + neutral sugar fraction, TSST-1 (BS) TSST-1 + basic sugar fraction. TSST-1, the percentages of CD3^+^ and CD45RA/CD62L + cells are shown. The number of DP positive cells (**c**); CD95 (**d**); CD45RO (**e**); CCR7 (**f**); and CD127 (**g**) is shown. Open bars, small lymphocyte-gated cells; closed bars, large lymphocyte-gated cells. n = 3. mean ± standard deviation (SD) is shown. * *p* < 0.05.

**Figure 2 microorganisms-12-00741-f002:**
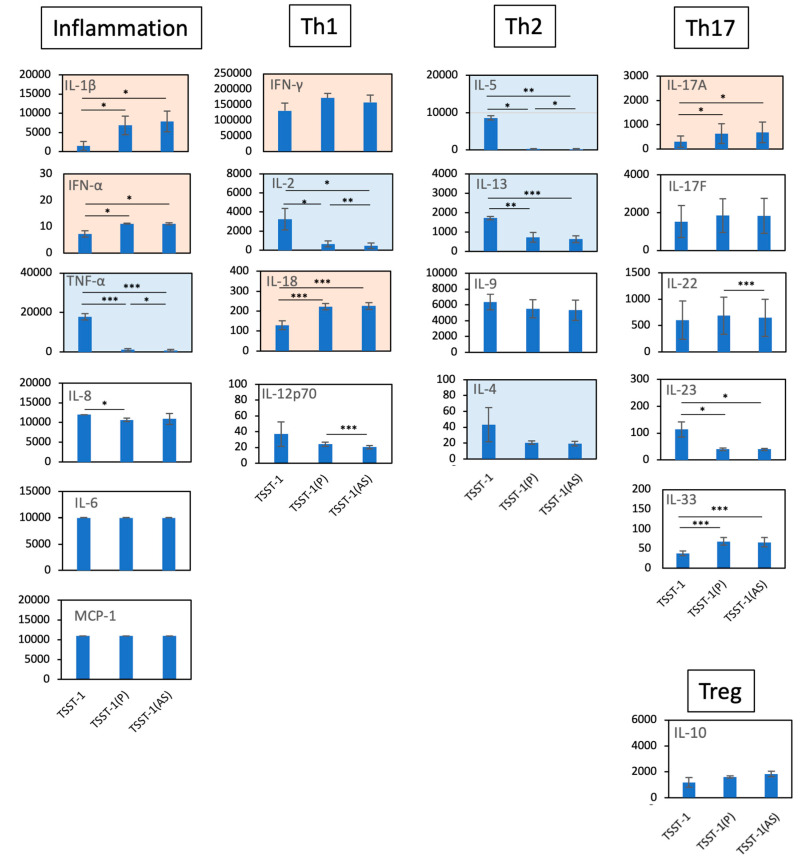
Effect of TSST-1 (P) and TSST-1 (AS) of C-KJ on the cytokine production of human PBMCs. Cytokine levels in the supernatants of TSST-1-stimulated PBMC in the presence of C-KJ components were measured using a LEGENDplex multi-channel cytokine evaluation kit. Inflammation; inflammation-related cytokines TNF-α and IL-6. Th1, Th1 cytokines IFN-γ and IL-2; Th2, Th2 cytokines IL-5, IL-13, IL-9, and IL-4; Th17, Th17 cytokines IL-17A, IL-17F, and IL-22; Treg, Treg cytokines IL-10. Pink panels, upregulated cytokines; blue panels, downregulated cytokines. n = 3. Data show mean ± SD. * *p* < 0.05, ** *p* < 0. 01, *** *p* < 0. 001.

**Figure 3 microorganisms-12-00741-f003:**
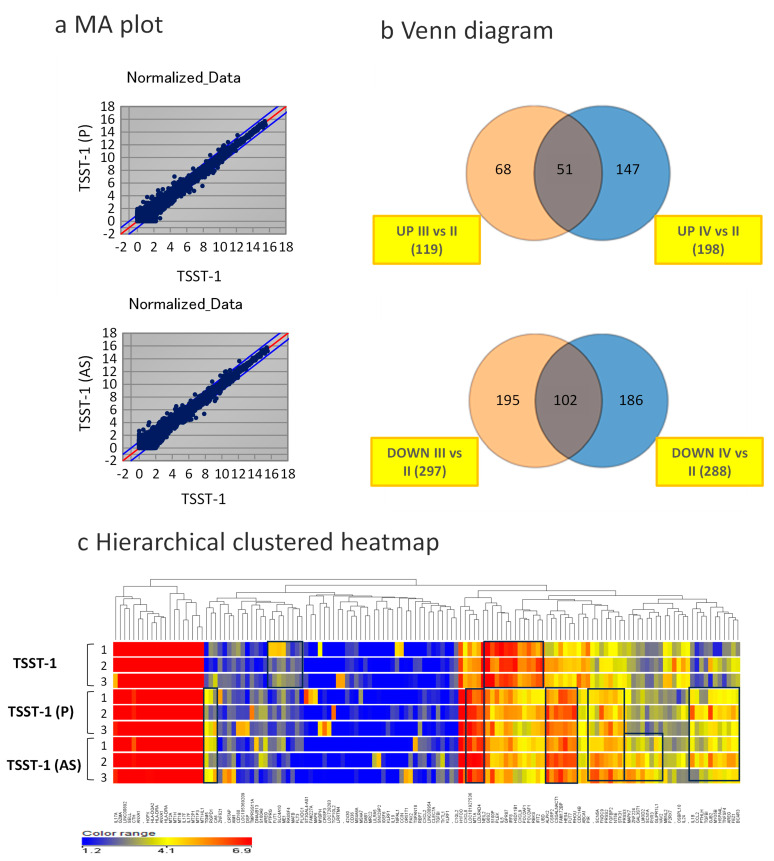
Analysis of differentially expressed genes (DEGs) based on the datasets obtained from TSST-1, TSST-1 (P) and TSST-1 (AS) T cells. Microarray analysis data (n = 3). (**a**) MA plots of TSST-1 vs. TSST-1 (P) (upper panel) and TSST-1 vs. TSST-1 (AS) (lower panel); Red line, 1:1; Blue lines, 1:2 or 2:1. (**b**) Venn diagram of increased DEGs of TSST-1 (P) (Orange) and TSST-1 (AS) (Blue) (upper panel) and decreased DEGs (lower panel). (**c**) Hierarchical clustered heatmap of TSST-1 (upper three columns), TSST-1 (P) (middle three columns), and TSST-1 (AS) (lower three columns). The squares show the clusters in which individuals in the same treatment (TSST-1/TSST-1 (P), TSST-1 (AS)) had distinctive similar scores.

**Figure 4 microorganisms-12-00741-f004:**
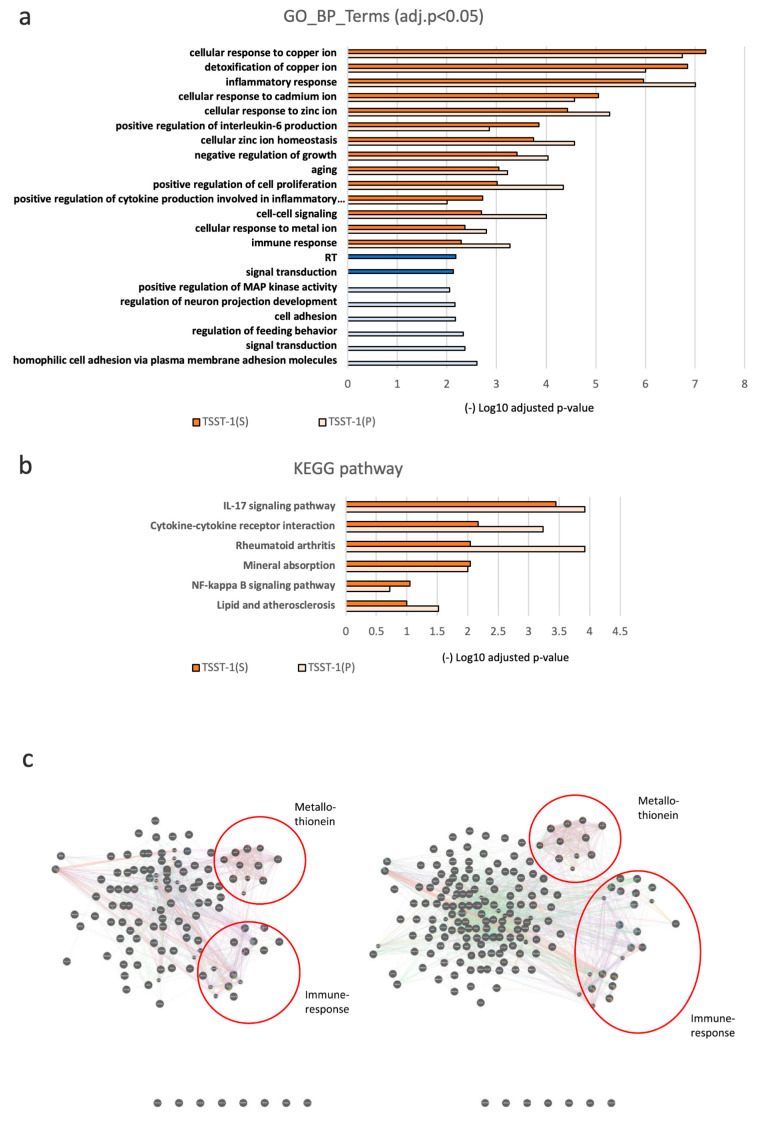
GO, KEGG, and protein–protein interaction networks analyses of DEGs. (**a**–**d**) GO (**a**); KEGG (**b**) pathway analysis; and protein–protein interaction analyses (**c**,**d**). In (**a**,**b**), the upregulated pathways are shown in orange, and the downregulated pathways are shown in blue [dark; TSST-1 (AS), pale; TSST-1 (P)]. (-) log_10_ adjusted *p*-values are shown. The results of the protein–protein interaction network analysis of TSST-1 vs. TSST-1 (P) (left panel) and TSST-1 vs. TSST-1 (AS) (right panel) are shown in (**c**). One prominent category is MT (upper panel), and the other is cytokines (lower panels). Two representative categories of IL-17 (lower left panel) and IL-1β (lower right panel) are shown in red circles. In (**d**), the protein-protein interactions in the prominent MT, IL-17, and IL-1β related pathways are shown. Left three figures shown under the label (protein); TSST-1 (P)/TSST-1, right three figures shown under the label (acidic sugar); TSST-1 (AS)/TSST-1. In (**c**,**d**), Red line, Physical Interactions; Purple line, Co-expression; Orange line, Predicted, Blue line, Co-localization; Green line, Genetic Interactions; Pale Blue line, Pathway; Pale Brown line, Shared protein domains.

**Figure 5 microorganisms-12-00741-f005:**
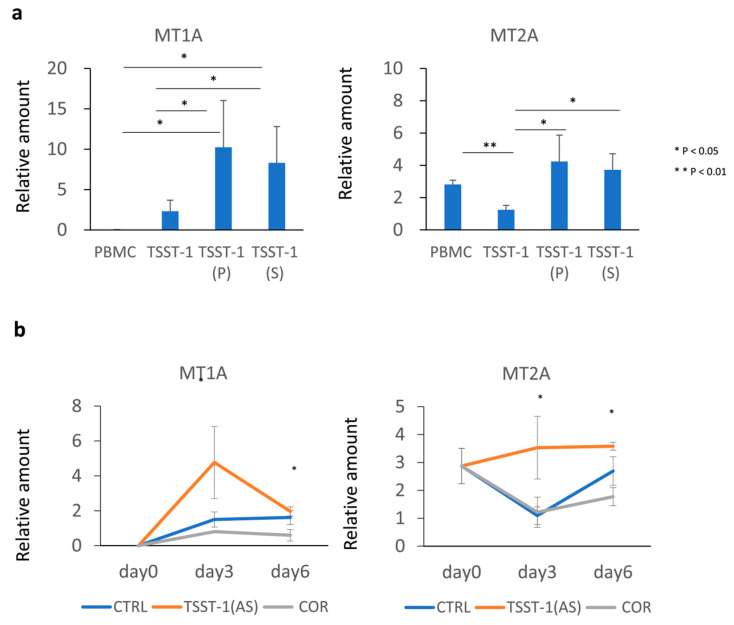
Effect of TSST-1 (P) and TSST-1 (AS) of C-KJ on the metallothionein (*MT*) mRNA production of human PBMCs. Effect of the C-KJ fractions on (**a**) *MT* expression and (**b**) kinetics of *MT* expression. The real-time PCR results for *MT1A* (left panel) and *MT2A* (right panel) are shown. In (**a**), vertical lines represent relative amounts of *MT* mRNA. In (**b**), blue line, TSST-1; red line, TSST-1 (AS); gray line; (COR). The samples were obtained on days 0, 3, and 6, and the mRNA levels were analyzed; (n = 3). Data show mean ± SD; * *p* < 0.05 and ** *p* < 0.01.

**Figure 6 microorganisms-12-00741-f006:**
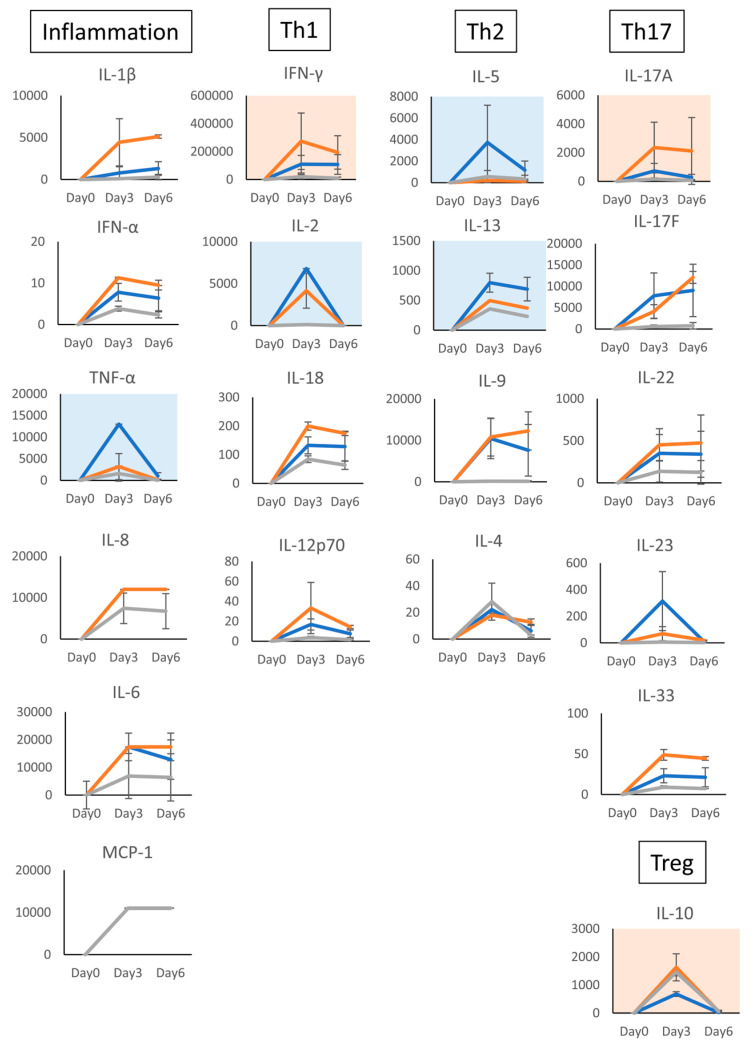
Effect of the acidic sugar (AS) fractions of *Coccomyxa* extracts and cortisol (COR) on the cytokine production of human T cells. Cytokines in the supernatants of TSST-1-stimulated PBMC cultured in the presence of AS or COR for 0–6 days were measured using the LEGENDplex multi-channel cytokine evaluation kit. Inflammation indicates inflammation-related cytokines, TNF-a, and IL-6. Th1: Th1 cytokines, IFN-γ and IL-2; Th2: Th2 cytokines, IL-5, IL-13, IL-9, and IL-4; Th17: Th17 cytokines, IL-17A, IL-17F, and IL-22; Treg, Treg cytokines, IL-10. Gray line, TSST-1; Orange line, TSST-1 (AS); Blue line, TSST-1 (COR). Pink panels, upregulated cytokines; blue panels, downregulated cytokines. n = 3; mean ± SD is shown.

**Figure 7 microorganisms-12-00741-f007:**
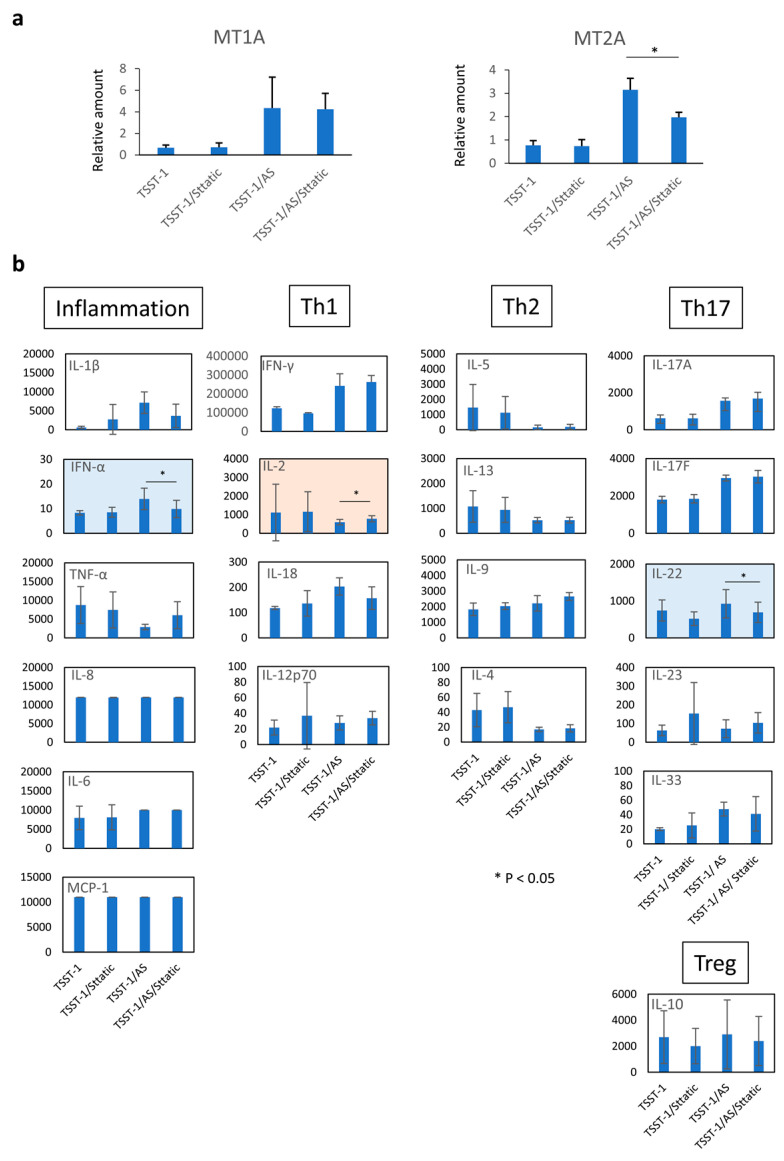
Effect of STAT-3 on the *MT* expression and cytokine production of human T cells. The real-time PCR results for *MT1A* (**left panel**) and *MT2A* (**right panel**) are shown. Vertical lines represent relative amounts of *MT* mRNA. The cytokines in the supernatants of TSST-1-stimulated PBMC cultured in the presence of AS (**a**) and STAT-3 inhibitor, static (**b**) as shown. Inflammation: inflammation-related cytokines, TNF-α, and IL-6; Th1, Th1 cytokines, IFN-γ and IL-2; Th2: Th2 cytokines, IL-5, IL-13, IL-9, and IL-4; Th17; Th17 cytokines, IL-17A, IL-17F, and IL-22; Treg: Treg cytokines, IL-10. Pink panels, upregulated cytokines; blue panels, downregulated cytokines; n = 3; data show mean ± SD is shown; * *p* < 0.05.

## Data Availability

The raw data supporting the conclusions of this article will be made available by the authors on request.

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
