# Peer review of "Coccomyxa subellipsoidea KJ Components Enhance the Expression of Metallothioneins and Th17 Cytokines during Human T Cell Activation"

_microorganisms, 2024, doi:10.3390/microorganisms12040741_

Round 1

Reviewer 1 Report

Comments and Suggestions for Authors

The manuscript "Coccomyxa KJ Components Enhance the Expression of 2 Metallothioneins and Th17 Cytokines During Human T Cell Activation 3" is a study like the previous work of the group with contradictory results, probably due to the fractionation of the algae extract. The manuscript focuses on protein and sugar extracts, whereas the previous study works with total extract (Reference 13: Microbiol Immunol. 2022 Aug;66(8):394-402). 

Why did the authors not include the previous extract in this paper?

Why were the neutral and basic fractions of C-KJ not tested in this study?

The experiment "Effect of TSST-1 (P) and TSST-1 (AS) of C-KJ on the cytokine production of human PBMCs" lacks control without stimulation of TSST-1. 

Figure S6 should be included in the main manuscript, but the legend needs to be improved.

Author Response

Response to the reviewers

We thank the reviewers for the thoughtful suggestions and insights, from which the manuscript has benefited. The manuscript has been checked and the necessary changes have been made in accordance with the reviewers’ suggestions. We have also added an explanation for Figure 1 and had our manuscript proofread by a professional English editing company. The responses to all comments are attached below.

Reviewer 1

The manuscript "Coccomyxa KJ Components Enhance theExpression of 2 Metallothioneins and Th17 Cytokines DuringHuman T Cell Activation 3" is a study like the previous work ofthe group with contradictory results, probably due to the fractionation of the algae extract. The manuscript focuses onprotein and sugar extracts, whereas the previous study workswith total extract (Reference 13: Microbiol Immunol. 2022Aug;66(8):394-402).

Why did the authors not include the previous extract in thispaper?

Reply: We appreciate the reviewer’s comment. Although we compared crude extract (CE) and fractionated samples, the previous text was insufficient. We aimed to narrow down the candidate molecules for immune regulation. Thus, we compared fractionated samples and CE for activation and Tscm markers because our previous results suggest that immune regulation involved an increase in Tscm-like cells. We thus decided to omit INT and focused on P and S. We have changed the text in L248-L260 as follows.

“The PBMCs were stimulated with TSST-1, an enterotoxin of Staphylococcus aureus (Ogston 1884) in the presence of crude extract (CE), P, S, and Int (at an equivalent of 300 μg/mL of CE). We divided small lymphocyte gates, indicating non-activated cells, and large lymphocyte gates, indicating activated cells, which are shown in Figure S2. The analysis revealed that most surviving human cells in each fraction after TSST-1 stimulation were T cells, and large lymphocyte-gated cells, which expressed the activation markers CD25 and PD-1, were increased in both CD4+ and CD8+ T-cells (Figure S2a, b). We compared the activation and differentiation markers of these T cells in detail. TSST-1-stimulated cells in the presence of P [TSST-1 (P)] and S [TSST-1 (S)] increased the ratio of CD45RA+CD62L+CD95+ Tscm , which was comparable with that of CE, while TSST-1 (Int) did not increase CD45RA+CD62L+CD95+ Tscm compared with TSST-1 alone (TSST-1) (Figures 1a and S3a). Therefore, we selected P and S as immune-modulatory fractions and performed further analyses.”

Why were the neutral and basic fractions of C-KJ not tested inthis study?

Reply: We appreciate the reviewer’s comment. The reason for why we did not analyze NS and BS might not have been mentioned clearly in the previous version of the manuscript. Therefore, we have revised the description in L260-L269 as below.

“After further fractionation of S, as shown in Figure S1, we compared AS, NS, and BS effects on T cell activation (Figures S2 c, d and S3b). No obvious difference was observed in the ratio of activated T cells among the sugar fractions, but TSST-1 (AS)-stimulated cells showed an increase in the Tscm ratio and the highest CD95 mean fluorescent intensity, while those in the cells treated with NS or BS showed no differences compared to TSST-1 (Figures 1b and S3b). Among the other Tscm markers, the expression of CD127 was increased in the large lymphocyte-gated Tscm fraction (Figure 1c–g). Tscm is a T cell subset with high potential to produce competent T cells. Our findings suggest that AS, containing the highest ratio of Tscm, induces and regulates T cell activation and differentiation; therefore, we selected AS and P for further analyses.”

We have also added CE data to Figure S2a and b. The previous data were moved to Figure S2c and d, and protein only data (data in the previous version of Figure 2c) were deleted.

The experiment "Effect of TSST-1 (P) and TSST-1 (AS) of C-KJ on the cytokine production of human PBMCs" lacks control without stimulation of TSST-1.

Reply: We appreciate the reviewer’s comment. As shown in Figure S2, almost none of the non-stimulated control cells were activated (CD24+PD-1+ cells) . Moreover, in a previous report (Oshima et al. 2022, Fig. S4), we checked the control without TSST-1 stimulation and confirmed that almost all cytokines examined were at the baseline level. Therefore, we omitted the related data in this figure for the sake of simplification. We believe that those data are not necessary to prove that P and AS similarly changed the profile of cytokine production.  

Figure S6 should be included in the main manuscript, but thelegend needs to be improved.

Reply: We appreciate the reviewer’s comment. We have included Figure. S6 in the main manuscript as Figure 4c and improved the figure legend as follows. (L323-L332)

“Figure 4. GO, KEGG, and protein–protein interaction networks analyses of DEGs. (a–d) GO (a); KEGG (b) pathway analysis; and protein–protein interaction network analyses (c) and (d). In (a) and (b), the upregulated pathways are shown in orange, and the downregulated pathways are shown in blue [dark; TSST-1 (AS), pale; TSST-1 (P)]. (-) log10 adjusted p-values are shown. The results of the protein–protein interaction network analysis of TSST-1 vs. TSST-1 (P) (left panel) and TSST-1 vs. TSST-1 (AS) (right panel) are shown in (c). One prominent category is MT (upper panel), and the other is cytokines (lower panels). Two representative categories of IL-17 (lower left panel) and IL-1β (lower right panel) are shown in red circles. In (d), the protein-protein interactions in the prominent MT, IL-17, and IL-1β related pathways are shown. Left three figures shown under the label (protein);  TSST-1 (P)/TSST-1, right three figures shown under the label (acidic sugar); TSST-1 (AS)/TSST-1.”

Reviewer 2 Report

Comments and Suggestions for Authors

The results of the study of green alga Coccomyxa subellipsoidea components for human T cell activation are presented. These data are very important for pharmacology, biotechnology, and phycology. The MS is clear and well-structured. The authors used proper methodology. The data interpreted appropriately and consistently throughout the manuscript. The results supported by detailed figures, but visibility of some illustrations should be improved.  It is necessary to note, that authors' suggestions confirm by statistical analysis. The conclusions of the paper are supported by data. Reference list contain the basis literature on the topic, but some of the reference are old.

Major suggestions:

1.         The MS dedicated to the study of Coccomyxa subellipsoidea, but in the title you have mentioned only Coccomyxa. I think, that you should correct Coccomyxa to Coccomyxa subellipsoidea.

2.         Add recent publication (2019-2023) in reference list. You have 44 reference, and about 15 published in 2019-2023.

Minor suggestions:

Line 22: I think, that it is better to replace the square brackets with a dash (correct “[acidic (AS), basic (BS), and neutral (NS)]” to  “- acidic (AS), basic (BS), and neutral (NS)”.

Line 41: Here you are talking about species Coccomyxa subellipsoidea, not a strain Coccomyxa subellipsoidea KJ. The information about the strain C-KJ; IPOD 41 FERM BP-22254 should be transfer to the section “Materials and Methods”.

Line 41: Correct “regulation[1,2]}” to “regulation [1,2]”. Pay attention to the space before square brackets.

Lines 41, 47, 49 and further:  It is necessary to specify the authors of the species and genus. 

Lines 221-239: I think, that some parts of these sections should be transferred to the “Materials and Methods”, including the protocol in Figure S1b.

Figure 1: Please, try to enlarge figures “a” and “b”.

Figure 3: The labels on figures “b” and “c” invisible, try to correct it.

Figure 4: Please, enlarge the labels on figure “a”.

Line 352: Modify “Because MT expression…”, because you have used the same words on line 344.

Lines 647-648: Correct reference 43.

Comments on the Quality of English Language

Minor editing of English language required

Author Response

Response to the reviewers

We thank the reviewers for the thoughtful suggestions and insights, from which the manuscript has benefited. The manuscript has been checked and the necessary changes have been made in accordance with the reviewers’ suggestions. We have also added an explanation for Figure 1 and had our manuscript proofread by a professional English editing company. The responses to all comments are attached below.

Reviewer 2

The results of the study of green alga Coccomyxa subellipsoideacomponents for human T cell activation are presented. Thesedata are very important for pharmacology, biotechnology, andphycology. The MS is clear and well-structured. The authors used proper methodology. The data interpreted appropriately and consistently throughout the manuscript. The results supported by detailed figures, but visibility of some illustrations should be improved. It is necessary to note, that authors' suggestions confirm by statistical analysis. The conclusions of the paper are supported by data. Reference list contain the basis literature on the topic, but some of the reference are old.

Major suggestions:

  1. The MS dedicated to the study of Coccomyxasubellipsoidea, but in the title you have mentioned onlyCoccomyxa. I think, that you should correct Coccomyxa to Coccomyxa subellipsoidea.

Reply: We appreciate the reviewer’s comment. According to the suggestion, we have changed the title to  “Coccomyxa subellipsoidea KJ components enhance the expression of metallothioneins and Th17 cytokines during human T cell activation.”

  1. Add recent publication (2019-2023) in reference list. You have 44 reference, and about 15 published in 2019-2023.

Reply; We appreciate the reviewer’s comment. Although most of the used references are not new, the number of studies related to coccomyxa and zinc is limited, and new evidences is scarce. However, we have added three new references.(L526, L527, 550)

  1. Mesas-Fernandez, A.; Bodner, E.; Hilke, F.J.; Meier, K.; Ghoreschi, K.; Solimani, F. Interleukin-21 in autoimmune and inflammatory skin diseases. Eur J Immunol 2023, 53, e2250075, doi:10.1002/eji.202250075.
  2. Fan, X.; Shu, P.; Wang, Y.; Ji, N.; Zhang, D. Interactions between neutrophils and T-helper 17 cells. Front Immunol 2023, 14, 1279837.
  3. Chen, B.; Yu, P.; Chan, W.N.; Xie, F.; Zhang, Y.; Liang, L.; Leung, K.T.; Lo, K.W.; Yu, J.; Tse, G.M.K. et al. Cellular zinc metabolism and zinc signaling: from biological functions to diseases and therapeutic targets. Signal Transduct Target Ther 2024, 9, 6, doi:10.1038/s41392-023-01679-y.

Minor suggestions:

Line 22: I think, that it is better to replace the square bracketswith a dash (correct “[acidic (AS), basic (BS), and neutral (NS)]”to “- acidic (AS), basic (BS), and neutral (NS)”.

Reply: We appreciate the reviewer’s comment. We have replaced the square brackets.(L21)

Line 41: Here you are talking about species Coccomyxa subellipsoidea, not a strain Coccomyxa subellipsoidea KJ. The information about the strain C-KJ; IPOD 41 FERM BP-22254should be transfer to the section “Materials and Methods”.

Reply: We appreciate the reviewer’s comment. We have moved the description to the “Materials and Methods.” in L114-L115

“The strain Coccomyxa sp. KJ (C-KJ; IPOD FERM BP-22254) was provided by Denso Co. Ltd. “

Line 41: Correct “regulation[1,2]}” to “regulation [1,2]”. Pay attention to the space before square brackets.

Reply: We appreciate the reviewer’s comment. We have added spaces between words and references.

Lines 41, 47, 49 and further: It is necessary to specify the authors of the species and genus.

Reply: We appreciate the reviewer’s comment. We have added the requested information to the manuscript. (L40, L43-L44, L48, L75-L76, L249)

Lines 221-239: I think, that some parts of these sections should be transferred to the “Materials and Methods”, including theprotocol in Figure S1b.

Reply: We appreciate the reviewer’s comment. We have moved some parts of the description to the methods in L116-L120.

“As shown in Figure S1a, the water-soluble extract of C-KJ was divided into two parts. One part was subjected to ultra-filtration to obtain the low-molecular-weight (MW; ≤ 3 kDa) compound (LWCO), and the other part was used for isolation of the protein (P) and sugar (S) fractions. The protocol for isolation, purification, and quantification of P fractions in C-KJ extracts is outlined in Figure S1b.”

Figure 1: Please, try to enlarge figures “a” and “b”.

Reply: We appreciate the reviewer’s comment. We have enlarged the panels.

Figure 3: The labels on figures “b” and “c” invisible, try to correctit.

Reply: We appreciate the reviewer’s comment. We have enlarged the labels.

Figure 4: Please, enlarge the labels on figure “a”.

Reply: We appreciate to the reviewer’s comment. We have enlarged the labels.

Line 352: Modify “Because MT expression…”, because you have used the same words on line 344.

Reply: We appreciate the reviewer’s comment. We have changed the description to “To confirm the induction of MTs and immune-related genes by the P and AS fractions,” in L367

Lines 647-648: Correct reference 43.

Reply: We appreciate the reviewer’s comment. We have changed the reference 43 as follows. (L503, L690-L691)

  1. Nishimura, N.; Reeve, V.E.; Nishimura, H.; Satoh, M.; Tohyama, C. Cutaneous metallothionein induction by ultraviolet B irradiation in interleukin-6 null mice. J Invest Dermatol 2000, 114, 343-348, doi:10.1046/j.1523-1747.2000.00862.x.

Comments on the

Minor editing of English language required

Reply: We have had our manuscript proofread by a professional English editing company.

Round 2

Reviewer 1 Report

Comments and Suggestions for Authors

All issues are well addressed and I think the manuscript is suitable to be accepted in its present form.

Author Response

I appreciate reviewer 1.